# Processing and Mechanical Properties of Basalt Fibre-Reinforced Thermoplastic Composites

**DOI:** 10.3390/polym14061220

**Published:** 2022-03-17

**Authors:** Xinying Deng, Ming Shun Hoo, Yi Wen Cheah, Le Quan Ngoc Tran

**Affiliations:** 1Singapore Institute of Manufacturing Technology, Agency for Science, Technology and Research (A*STAR), Singapore 138634, Singapore; hoom0001@e.ntu.edu.sg (M.S.H.); cheah_yi_wen@simtech.a-star.edu.sg (Y.W.C.); tranlqn@simtech.a-star.edu.sg (L.Q.N.T.); 2School of Materials Science and Engineering, Nanyang Technological University, Singapore 639798, Singapore

**Keywords:** thermoplastic composite processing, basalt, mechanical properties

## Abstract

Basalt fibre is derived from volcanic rocks and has similar mechanical properties as glass fibre. However, poor fibre-matrix compatibility and processing issues are the main factors that have restricted the mechanical performance of basalt fibre-reinforced thermoplastic composites (BFRTP). In this work, basalt continuous fibre composites with polypropylene (PP) and polycarbonate (PC) matrices were studied. The composites were processed by compression moulding, and a processing study was conducted to achieve good quality composites. For the BF-PC composites, the optimisation of material preparation and processing steps allowed the polymer to impregnate the fibres with minimal fibre movements, hence improving impregnation and mechanical properties. For BF-PP composites, a compatibiliser was required to improve fibre-matrix compatibility. The compatibiliser significantly improved the tensile and impact strength values for short BF-PP composites and continued to increase at 40 wt%. Furthermore, the analytical modelling of the Young’s moduli indicated that the induced fibre orientation during processing for short BF-PP composites and unidirectional (UD) BF-PC composites had better stress transfer than that of UD BF-PP composites.

## 1. Introduction

Basalt fibre is a bio-derived mineral fibre from volcanic rocks and has several advantages, such as good chemical resistance and mechanical properties [1,2,3,4,5,6]. Basalt fibre has similar tensile properties as glass fibres and has a higher maximum service temperature than glass and carbon fibres, as shown in Table 1. This enables basalt fibre composites to be a viable and sustainable alternative to glass fibre composites for structural applications. Moreover, it has a higher maximum service temperature and good chemical resistance, which makes it suitable for applications in harsh environments, such as composite pipes for the oil and gas industries or chemical storage tanks [1].

Basalt fibre has been studied by several scientists to explore the fibre used as reinforcement for polymer composites [2,3,7,8,9,10,11]. With good mechanical properties in combination with high temperature and alkaline resistance, basalt fibre is a good candidate for composites with thermoset matrices such as epoxy and polyester. Arshad et al. [7] reported epoxy hybrid composites having a high resistance to changes in temperature and significant enhanced mechanical performance. Basalt fibre reinforced epoxy composites also demonstrated higher resistance to aging in alkaline medium and heat than those of glass fibre epoxy composites based on E-glass and S-glass fibres, while the mechanical properties of the basalt fibre composites are closer to those that of S-glass composites and greater than those of E-glass composites. These properties are connected to the high adhesion between the basalt fibres and epoxy [8].

With increasing demand for lightweight, sustainable materials for metal-replacement in various industries, the global thermoplastic composite market has been projected to reach USD 16.3 billion by 2023 [12]. Thermoplastic composites can be recycled and welded, allowing composites to be part of a circular economy and also offer the possibility of forming large composite structures in a shorter timeframe [13,14,15]. However, the research for thermoplastic composites is less established than that for thermoset composites due to issues such as fibre-matrix compatibility, high processing temperatures, and processing difficulty [13,14,15,16]. Regarding basalt fibre reinforced thermoplastics BFRTP), most of studies have been focused on short basalt fibre composites with thermoplastic matrices including polypropylene (PP), polypropylene/poly(butylene terephthalate) and poly(vinylidene fluoride)/poly(methyl methacrylate) blends for manufacturing with injection moulding process [2,3,4,16,17,18]. The mechanical properties of the short-fibre composites were limited by the loading of the fibres, in which the fibre loading was recommended at lower than 20% wt. There are also limited fundamental studies to understand the potential of basalt fibre thermoplastic composites by looking into the performance of unidirectional continuous fibre composites for better design for application of the composites. 

Other aspects include fibre-matrix compatibility and adhesion. The sizing of commercial continuous basalt fibre was developed for epoxy applications, limiting its mechanical performance. Attempts have been made to improve the fibre-matrix adhesion through the use of different compatibilisers or sizing agents [18,19]. Russo et al. [19] showed that the use of polypropylene-graft-maleic anhydride improved the flexural and impact performance of continuous BF-PP composites. However, the highest flexural strength value achieved was 81.1 MPa, which was significantly lower than the expected strength values of continuous glass fibre thermoplastic composites. This could be due to lower fibre volume fraction or composite quality issues, which were not investigated in the study. Therefore, there is a need to understand the processing of the continuous basalt fibre thermoplastic composites in order to achieve a good quality composite with the consistent and good mechanical performance required for structural applications.

In the present study, a processing study was conducted for the continuous basalt fibre thermoplastic composites to resolve fibre spreading issues and improve matrix impregnation. Basalt fibre is typically sized for epoxy polymers; therefore, polypropylene-graft-maleic anhydride was used as a compatibiliser to improve the fibre-matrix adhesion for basalt fibre-polypropylene (BF-PP) composites. Tensile properties, degree of impregnation, and fibre volume fraction were characterised. Additionally, the effect of compatibiliser on the tensile and impact properties of short BF-PP composites was also studied. Analytical models were used to predict the Young’s modulus values for the short and continuous fibre composites, and the predictions will be compared to the experimental results. 

## 2. Materials and Methods

### 2.1. Materials

#### 2.1.1. Fibre

The basalt fibre used in this study was in the form of unidirectional (UD) basalt fabric, provided by Sure New Material (Zhejiang) Co. Ltd. (Tongxiang, China). The UD fabric had a linear density of 400 Tex and a thickness of 0.243 mm. The short basalt fibre was cut from this fabric to approximately 10 mm length before further processing. 

#### 2.1.2. Polymer

Commercial grade polycarbonate (PC) film supplied by Pacco Chemical (S) Pte Ltd. (Singapore) was used in this study. The PC film has average thickness of 0.1 mm and density of 1.2 g/cm^3^.

Polypropylene (PP) pellets, Cosmoplene, Y101E grade, were provided by The Polyolefin Company Pte. Ltd. (Singapore) and were used in compounding process and to make the PP film using the press at 250 °C. The PP film had average thickness of 0.1 mm to 0.2 mm and density of 0.90 g/cm^3^. 

#### 2.1.3. Compatibiliser

Polypropylene-graft-maleic anhydride (MAPP) pellets were utilised to improve fibre-matrix adhesion in basalt fibre reinforced PP composites, which were supplied by Sigma-Aldrich, (St. Louis, MO, USA). The MAPP has 8 wt% to 10 wt% of grafted maleic anhydride, melting point of 156 °C and density of 0.934 g/mL at 25 °C.

### 2.2. Composite Processing

#### 2.2.1. Processing of UD Basalt Fibre Composites by Compression Moulding

UD basalt fibre composites with both PC and PP were prepared using a stack of fibre fabrics and polymer films under compression moulding process. Layers of film and basalt fabric were cut and stacked in alternate sequence before being placed in a rectangular picture frame mould with inner dimensions of 250 mm by 150 mm or a square picture frame mould with inner dimensions of 200 mm by 200 mm. The basalt fabric and PP film were used without any pretreatment, but PC film was dried overnight at 80 °C. The compression moulding process was conducted using the P500 Collin hot press. In general, the composites went through a pre-consolidation phase to allow time for the matrix to impregnate the fibres, before going into the consolidation phase, cooled down to room temperature and followed by composite demoulding. The parameters for the pre-consolidation and consolidation phases for the BF-PC and BF-PP composites were summarised in Table 2. In this paper, the parameters for the consolidation phase were kept constant for most of the samples except for two BF-PC samples. 

PP was provided as pellets and was processed into films using hot pressing process. Hot press, 500P, from Collin (Ebersberg, Germany) was used to make the PP film (with and without compatibiliser) at 250 °C. For the PP film with compatibiliser, prior to the making of the film, a compounding process was required. 3 wt% of MAPP was pre-mixed with PP pellets before the compounding process. Compounding of the materials was conducted using a twin-screw extruder, Micro 27, from Leistritz (Nuremberg, Germany), at 100 rpm with a temperature profile ranging from 150 °C to 170 °C.

#### 2.2.2. Short Fibre Composite Processing 

BF-PP short-fibre composites were compounded at fibre loadings of 20 wt%, 30 wt%, and 40 wt%. The BF and PP pellets were used as received and appropriate amount of BF and PP pellets were weighed and pre-mixed before compounding process. Compounding of the materials was carried out using a lab-scale twin-screw extruder, HAAKE^TM^ Minilab 2, from Thermo Fisher Scientific (Waltham, MA, USA), at 200 °C and 65 rpm. The compounded samples were then injection moulded using a lab-scale injection moulding machine, HAAKE^TM^ Minijet, from Thermo Fisher Scientific (Waltham, MA, USA). An injection temperature of 200 °C, injection pressure of 150 bar for 20 s and post pressure of 100 bar post pressure for 10 s were used to obtain the specimens required for tensile and impact tests. 

### 2.3. Characterisation Methods

#### 2.3.1. Polymer Flow Measurement 

Melt flow index (MFI) of the studied polymers were determined for setting process conditions of the composites. The MFI was characterised using the melt flow tester, CEAST MF20, from Instron (Turin, Italy) and in accordance with ASTM D1238 [20]. The MFI analysis for PP was conducted at temperatures from 170 °C, 190 °C, 210 °C, 230 °C, and 250 °C with a 2.16 kg weight. The MFI versus temperature graph for PP was then plotted to determine the relationship between the MFI and temperature of the PP matrix. The MFI analysis of PC matrix was conducted at temperatures of 220 °C, 240 °C, 260 °C, 280 °C, and 300 °C with a 1.2 kg weight. 

#### 2.3.2. Measurement of Fibre Volume Fraction of Composites

The fibre volume fraction of the continuous and short-fibre composites was measured using the thermogravimetric analyser (TGA), TGA Q500, from TA instruments (New Castle, DE, USA). 10 mg to 20 mg of each specimen were tested from room temperature to 800 °C at a ramp rate of 10 °C per min. At least three tests were conducted for each sample. 

Analysis was conducted using TA Universal Analysis (New Castle, DE, USA). As BF do not burn off in air even at 800 °C, though PP burns off cleanly, the remaining weight fraction at 800 °C correspond to the weight fraction of the BF in BF-PP composites. 

#### 2.3.3. Mechanical Tests

The tensile tests were conducted using the Universal Testing Machine 5982 from Instron (Norwood, MA, USA), with non-contacting video extensometer, AVE, for strain measurement, and in accordance with ASTM D3039 [21] and ISO 527 [22] for continuous fibre composites, and polymer and short-fibre composites, respectively. A minimum of 5 specimens were tested for each sample. For the continuous fibre composites, the nominal specimen dimension was 150 mm by 15 mm by 1 mm or 2 mm and emery cloth was used to improve the grip during testing. A cross head speed of 2 mm/min was used. For the polymer and short-fibre composites, the specimen type used was 1BA. A crosshead speed of 5 mm/min was used for PP polymer and 2 mm/min was used for the short-fibre composites. 

Flexural test was performed using the Universal Testing Machine 5982 from Instron (Norwood, MA, USA)in accordance with ASTM D790 [23] and a minimum of 5 specimens were tested per sample. The nominal specimen dimension was 60 mm by 12.7 mm by 1 mm and a span of 26 mm was used. Flexural test was not reported for BF-PP composite as the samples did not break. 

Izod impact test was conducted using Pendulum Impact Tester, HIT25P from ZwickRoell (Ulm, Germany) and in accordance with ASTM D256 [24]. The specimens were notched and had a nominal dimension of 63.5 mm by 12.7 mm by 3 mm. A minimum of 10 specimens were tested for each sample. 

#### 2.3.4. Investigation of Composite Quality by Microscopic and FESEM Images

The quality of produced composites was characterised using microscopic images of their cross-sections. The composite cross-sections were cut and mounted in a resin block. A surface preparation process including grinding and polishing was conducted for the mounted composite samples to achieve clean surfaces that were ready for analysis under a microscope.

Microscopic analysis of the mounted composites was performed using GX51 inverted optical microscope with DP72 attachment from Olympus (Tokyo, Japan), and AnalysisPro software by Olympus (Tokyo, Japan). 

For analysis under field emission scanning electron microscope (FESEM), Gold Sputter Coater, EM ACE200, from Leica (Vienna, Austria) was used to coat the surface of the samples before imaging using the FESEM. The samples are secured to the rotating holder of the sputter holder using carbon tape and undergo platinum sputtering for 60 s. After sputter coating the samples, the samples are secured to the sample holder of the FESEM and taped with copper tape at the corners to increase sample conductivity. The samples were then imaged using the FESEM, Ultra Plus from Carl Zeiss (Oberkochen, Germany) using different magnifications to observe differences in the fibre-matrix interactions of each sample.

### 2.4. Sample Annotation

The samples in this paper are described in the following manner, as shown in Table 3. For example, the BF-PC composite that was processed at pre-consolidation temperature of 170 °C with no variation in consolidation parameters, i.e., 220 °C and 5 bar, is denoted as ‘PC-170’. The BF-PC composite that was processed at pre-consolidation temperature of 170 °C and with variation in consolidation temperature, i.e., 240 °C instead of 220 °C, is denoted as ‘PC-170, 240’.

## 3. Results

### 3.1. Process Optimisation for UD Basalt Fibre PC Composites

Regarding UD composite preparation using compression moulding, significant fibre movements occurred when high temperature or pressure were applied, but poor composite quality resulting from insufficient consolidation was obtained if low processing pressure and temperature were used. In a previous study on the compression moulding of carbon fibre-polycarbonate composites, it was found that the addition of a pre-consolidation step at 170 °C at 0.2 bar prevented extensive fibre movement [25]. Moreover, a consolidation temperature of 220 °C reduced fibre spreading as compared to a consolidation temperature of 240 °C, while an MFI greater than 20 g/min would aid in matrix impregnation. Hence, the initial process optimisation was carried out with the pre-consolidation phase (170 °C, 0.2 bar, 2 to 5 min), followed by the consolidation phase. Such a mild consolidation procedure resulted in poor composite quality, and hence, the processing steps were modified, using a higher pre-consolidation temperature to improve the impregnation and a lower consolidation temperature to prevent excessive fibre movement, as displayed in Figure 1. In addition, the pre-consolidation temperatures of 260 °C and 280 °C were selected, based on the MFI of PC film, to be close to a MFI of 20 g/min as presented in Figure 2.

There was a significant number of voids, particularly between the fibre bundles for the undried BF-PC composite processed at a lower pre-consolidation temperature, as highlighted in the circles in Figure 3a. Higher pre-consolidation temperature and drying significantly improved the impregnation and matrix was observed in the middle of the fibre bundle, as seen in the highlighted areas in Figure 3b,c. The improved impregnation was also reflected in the increases in tensile and flexural properties (Table 4). Improved impregnation indicated fewer voids, as seen in Figure 3b,c, and voids are stress concentrators, which would have caused premature failure during mechanical testing.

In addition, it was verified that the consolidation temperature of 220 °C and pressure of 5 bar are the optimised consolidation parameters, as varying the consolidation temperature and pressure did not yield better tensile properties, as shown in Table 4. A further increase in the consolidation parameters could possibly improve the degree of impregnation, but it could also cause fibre spreading issues and hence, the tensile properties did not improve with higher temperature or pressure. 

The investigation of composite quality was also conducted using FESEM images, as shown in Figure 4. Polymer residues were observed on the fibres of BF-PC composites processed at different pre-consolidation temperatures, indicating good fibre-matrix adhesion, as seen in Figure 4a–c. However, some slight differences were observed for the fibre tracks where the fibre was pulled out of the matrix. At a lower pre-consolidation temperature of 170 °C and with undried polymer, the fibre track was smoother with minimal matrix deformation, whereby at higher pre-consolidation temperatures of 260 °C and 280 °C, more extensive matrix deformation with multiple steps/holes was observed on the fibre tracks. This indicated slightly improved fibre-matrix adhesion with drying and with higher pre-consolidation temperatures, which aided in stress transfer from matrix to fibre and improved the mechanical properties. 

### 3.2. Processing Optimisation for UD Basalt Fibre PP Composites

The melt flow index of the PP polymer at different temperatures was measured, as presented in Figure 5a and hence, the pre-consolidation temperatures of 210 °C and 250 °C were selected in consideration of the corresponding MFI value of around 20 g/10 min. The optical microscope images showed that the BF-PP composites seemed to have a lower fibre volume fraction as compared to BF-PC composites and had a similarly moderate level of impregnation despite using different pre-consolidation temperatures, as seen in Figure 6a–c. There were matrix and small voids observed in the fibre bundles, indicating that the viscosity of the PP polymer was low and could flow into the fibre bundle. Hence, the change in pre-consolidation temperature did not have an effect on the degree of impregnation for BF-PP composites. Similarly, the tensile properties did not have significant changes with the change in pre-consolidation temperature, as seen in Table 5. However, it was noted that a higher pre-consolidation temperatures increase polymer flow, and hence, the thickness of the BF-PP composites decreased from 2.0 mm to 1.6 mm with an increase in the pre-consolidation temperature from 170 °C to 250 °C. 

Figure 5 presents the MFI of PP with and without compatibiliser. The MFI of PP with compatibiliser is higher than that of PP without compatibiliser. However, there was no significant improvement in the degree of impregnation as seen from the cross-section presented in Figure 6a,d. Thus, the improved tensile properties for the BF-PP composite with compatibiliser are attributed to the improved fibre-matrix interface as observed in FESEM images in Figure 7a,b. A smooth fibre surface could be seen in unmodified BF-PP, whereas polymer residues were observed when MAPP was added. 

### 3.3. Processing Conditions and Mechanical Properties of Short Basalt Fibre PP Composites

The tensile and impact properties of the basalt composites were measured and presented in Figure 8a,b and Table 6. The tensile and impact strength for BF-PP short-fibre composites did not increase beyond a fibre loading of 30 wt% when no compatibiliser was used. This might be attributed to the presence of too many fibre ends within the composite, which could have resulted in crack initiation and potentially the composite’s failure [26]. On the other hand, when compatibiliser was added, the tensile and impact strength values were significantly higher than those without compatibiliser and continued to increase at fibre loading of 40 wt%. It was noted that at a higher fibre loading, there is more yielding behaviour for the BF-PP composites with compatibiliser, which is reflected in the lower measured values for Young’s modulus for the BF-PP composite with compatibiliser. 

The improvement in tensile and impact strength of the composites with the use of compatibiliser can be attributed to the improved fibre-matrix interface between the basalt fibre and the PP matrix, leading to better stress transfer. This could be observed from the FESEM images of the composite fractured surface, as seen in Figure 9a,b. There were a higher number of holes and large gaps observed between the fibres and matrix for unmodified BF-PP, while the gaps were significantly smaller when MAPP was added to the polymer matrix. In addition, a smooth fibre surface could be seen for the unmodified BF-PP, whereas a slightly rougher fibre surface with polymer residues was observed in the modified composite system.

## 4. Discussion

### 4.1. Analytical Models for Modelling of Young’s Modulus

The modulus of polymers and composites can be predicted using models such as the rule of mixture [27] and the Halpin–Tsai model [28]. However, it should be noted that these models often over predict the modulus of polymers and composites since they were originally developed for unidirectional composites and assumed a perfect fibre-matrix interface, and modifications are required. 

#### 4.1.1. Halpin–Tsai Model

The Halpin–Tsai model [28] is described below as follows:(1)$$E=E_{m}\left(\frac{1+{\eta \xi v}_{f}}{1-{\eta v}_{f}}\right)$$
(2)$$\eta =\frac{\frac{E_{p}}{E_{m}}-1}{\frac{E_{p}}{E_{m}}+\xi }$$

Where E_p_ is the modulus of the modifier (short fibre in this study), E_m_ is the modulus of the matrix, and v_f_ is the volume fraction of the modifier. The shape factor (ξ) was suggested to be twice the aspect ratio of the modifier. The aspect ratio (AR) is calculated by dividing the length, l_p_, by twice the radius, r_p_. However, the Halpin–Tsai model often over predicts the stiffness and is unable to take into account the orientation of the modifiers. Van Es [29] proposed a change in the shape factor expression to account for the random orientation of rod-like modifiers, such that the modulus of the modified polymer or composite is given by the following:(3)$$E=0.184{\;E}_{//}+0.816{\;E}_{T}$$
where the modulus parallel to the loading direction (E_//_) and the modulus transverse to the loading direction (E_T_) moduli are calculated using Equation (1) with ξ = 2 for E_T_ and ξ = (0.5AR)1.8 for E_//_. This will be referred to as the Halpin–Tsai (random) model.

#### 4.1.2. Rule of Mixtures

Fibre reinforced materials that possess high mechanical properties are typically based upon carbon and glass fibres, and the reinforcement effect depends on the length, distribution, orientation, type, processing, and interfacial compatibility of the fibres.

The elastic modulus of the composite system can be derived from the rule of mixtures (ROM) and is given by the following [27]:(4)$$E=E_{F}v_{F}+E_{m}v_{m}$$
where E_F_ and E_m_ are the elastic modulus of the fibre and the matrix, respectively, and v_F_ and v_m_ are the volume fraction of the fibre and matrix, respectively. 

### 4.2. Young Modulus of Short Basalt Fibre PP Composites

Young’s moduli of the short BF-PP composites were predicted using the Halpin–Tsai models. The parameters used are summarised in Table 7. The experimental results match better with the Halpin–Tsai model as compared to the Halpin–Tsai (random) model, indicating that there is some degree of orientation of the fibres along the testing direction, see Figure 10. This could be due to the induced polymer flow during the extrusion and injection moulding processes, which allows the fibres to be aligned along the length of the tensile specimens. In addition, the measured dimension of the basalt fibre has a large standard deviation that contributed to some over-prediction. 

### 4.3. Young Modulus of UD Basalt Fibre Composites

The Young’s moduli of the UD composites were predicted using the rule of mixtures and the parameters used were the same as those of the BF-PP short-fibre composite. The density and Young’s modulus of polycarbonate were taken to be 1.2 g/cm^3^ and 2.3 GPa [31], and the parameters used for basalt fibre and polypropylene were summarised in Table 7. 

The experimental Young’s modulus values reached at least 69% and 81% of the theoretical values for BF-PP and BF-PC composites, respectively, as shown in Table 8. This indicated BF-PC composites had better stress transfer than BF-PP composites. It was noted that the efficiency factor for BF-PP composites did not differ with the use of the compatibiliser, which may be due to the reduction of Young’s modulus of the matrix by the compatibiliser that would affect the value of the theoretical Young’s modulus. This could be further enhanced by refinement of the pre-consolidation process parameters and with further advancement in compatibiliser technology. 

### 4.4. Tensile Strength of Basalt Fibre Composites

The tensile strengths of the BF-PC and BF-PP composites in this study were plotted against the tensile strengths of BF and GF composites in different matrices, which were reported by other researchers, as shown in Figure 11. As compared to BF and glass fibre (GF) thermoset composites, the studied BF-PC thermoplastic composite could achieve comparable mechanical properties. These BFRTP could be a bio-derived alternative to GF composites for structural applications that can be adopted by various sectors such as construction, automotive, and marine applications. 

## 5. Conclusions

Basalt continuous fibre composites with Polypropylene (PP) and Polycarbonate (PC) matrices were fabricated using a compression moulding process and process optimisation was conducted to resolve the fibre spreading issues and improve fibre-matrix impregnation. Drying is critical for BF-PC composites and the modified procedure with a higher pre-consolidation temperature enabled good matrix impregnation into the basalt fibres with minimal fibre movements. For BF-PP composites, a compatibiliser was required to improve fibre-matrix compatibility and improve the tensile strength. Moreover, it was found that the tensile and impact properties for BF-PP short-fibre composites did not increase beyond a fibre loading of 30 wt% if no compatibiliser was used, but the values continued to increase at 40 wt% with the use of compatibiliser. When benchmarked against BF and glass fibre (GF) thermoset composites, a lower fibre volume fraction was obtained for the BFRTP, but the BF-PC composites could achieve comparable mechanical properties. Analytical modelling of the Young’s moduli of the BF-PP short-fibre composites was predicted with the Halpin-Tsai model as compared to the Halpin-Tsai (random) model, indicating induced fibre orientation along the testing direction during processing for short BF-PP composites. The experimental Young’s modulus values reached at least 69% and 81% of the theoretical values for BF-PP and BF-PC composites, respectively, which indicated that BF-PC composites had better stress transfer than BF-PP composites. This study lays the foundation for the processing and basic mechanical properties of basalt fibre thermoplastic composites with PC and PP as the polymer matrix and contributes to the working knowledge of BFRTP for potential future applications in various sectors such as construction, aerospace, and marine. 

## Figures and Tables

**Figure 1 polymers-14-01220-f001:**
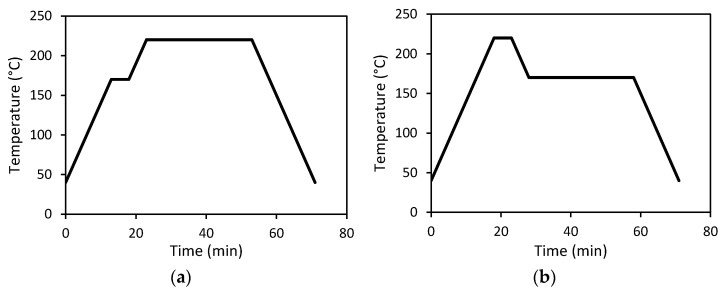
Illustration of (**a**) the conventional compression moulding process and (**b**) the modified compression moulding process.

**Figure 2 polymers-14-01220-f002:**
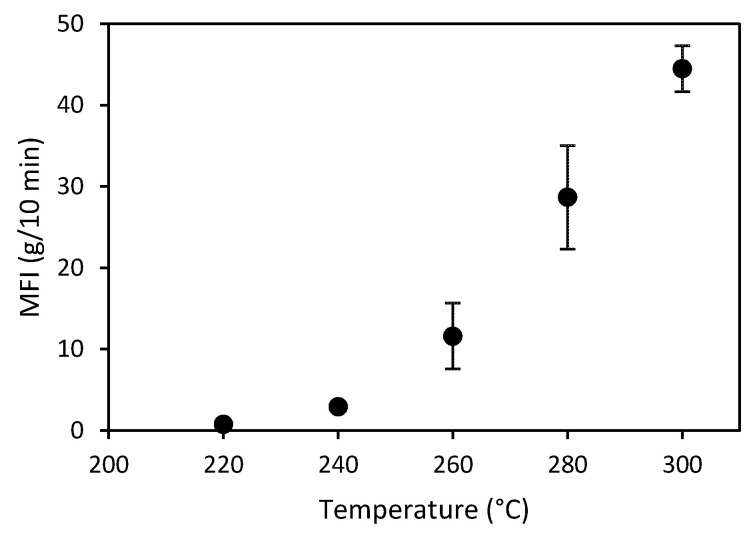
MFI of polycarbonate film against temperature.

**Figure 3 polymers-14-01220-f003:**
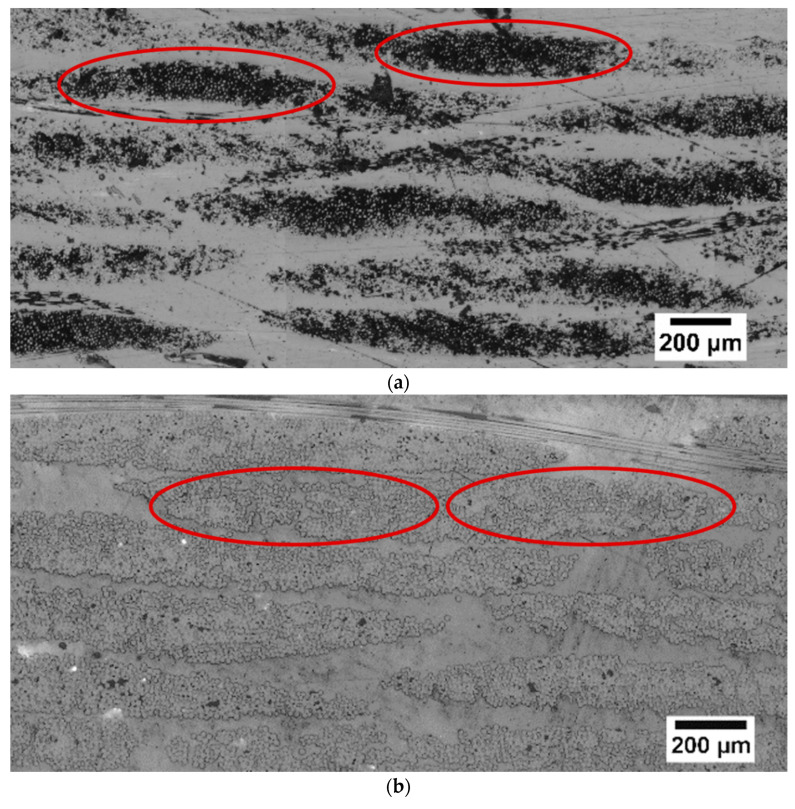
Varying degrees of impregnation observed through optical microscope images of BF-PC composites processed at pre-consolidation temperatures of (**a**) 170 °C (u), (**b**) 260 °C, and (**c**) 280 °C.

**Figure 4 polymers-14-01220-f004:**
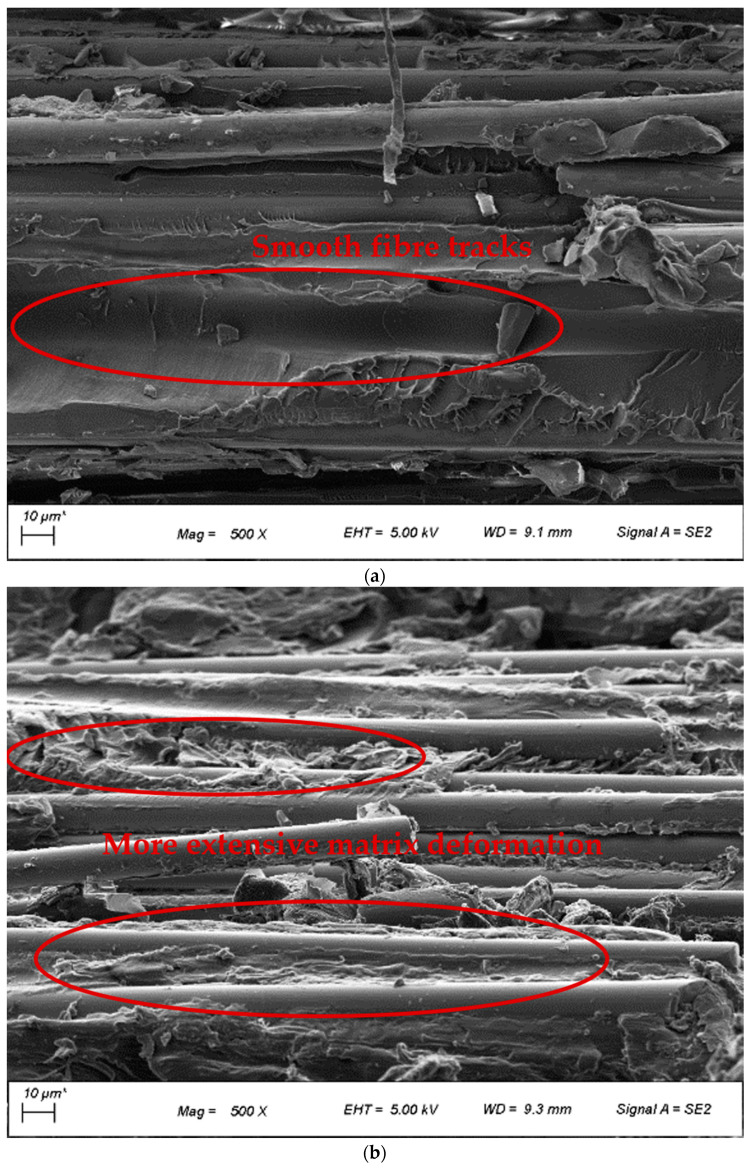
FESEM images of BF-PC composites processed at pre-consolidation temperatures of (**a**) 170 °C (u), (**b**) 260 °C, and (**c**) 280 °C.

**Figure 5 polymers-14-01220-f005:**
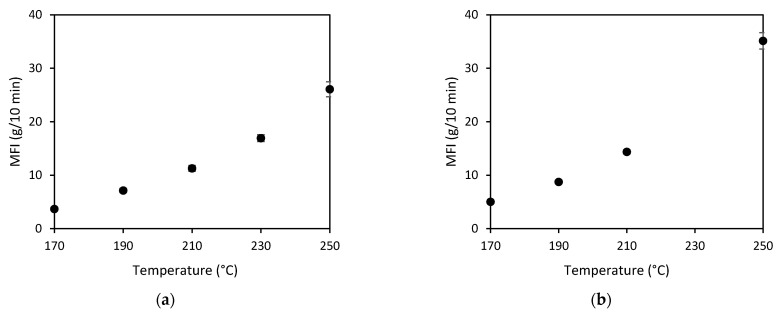
MFI against temperature for (**a**) PP and (**b**) PP with 3 wt% MAPP.

**Figure 6 polymers-14-01220-f006:**
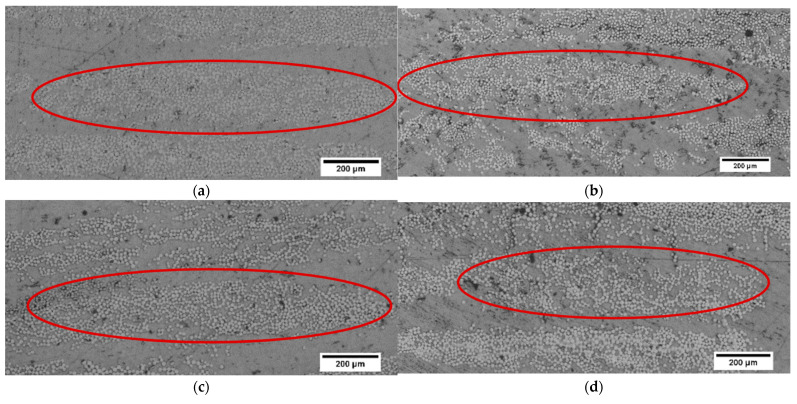
Similar degree of impregnation observed through optical microscope images of BF-PP composites processed at pre-consolidation temperatures of (**a**) 170 °C, (**b**) 210 °C, and (**c**) 250 °C, and (**d**) BF-MAPP composite processed at pre-consolidation temperature of 170 °C.

**Figure 7 polymers-14-01220-f007:**
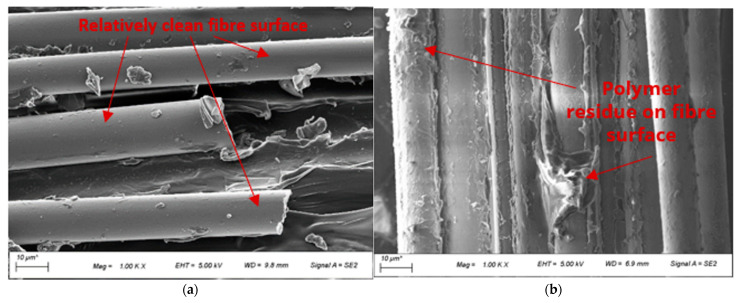
FESEM images of the BF-PP composites processed at the same conditions and (**a**) without compatibiliser and (**b**) with compatibiliser.

**Figure 8 polymers-14-01220-f008:**
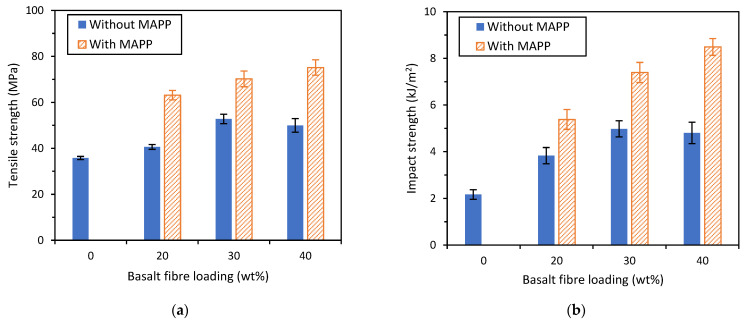
(**a**) Tensile strength and (**b**) impact strength of short BF-PP composites with and without the use of MAPP as compatibiliser.

**Figure 9 polymers-14-01220-f009:**
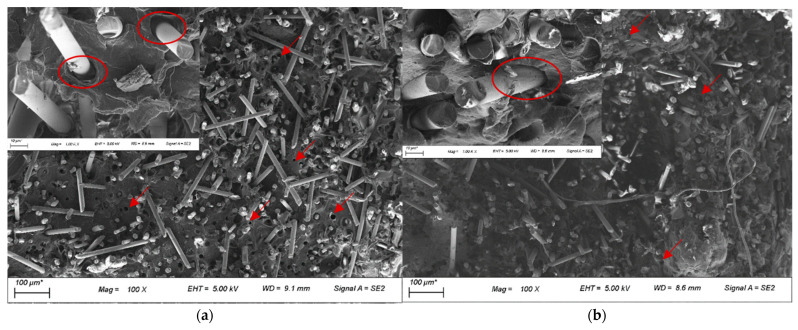
FESEM images with inserts at higher magnification of the short BF-PP composites (**a**) without compatibiliser and (**b**) with compatibiliser. Note: Fibre-matrix gap highlighted by red circle and holes highlighted by red arrows.

**Figure 10 polymers-14-01220-f010:**
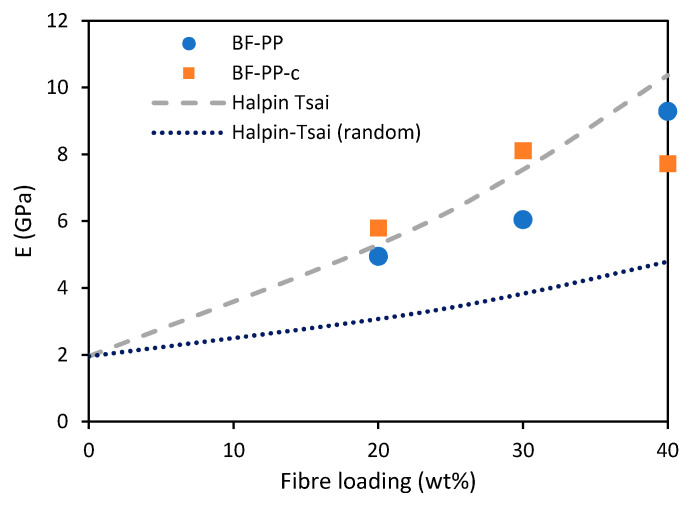
Predicted and measured Young’s moduli of short BF-PP composites.

**Figure 11 polymers-14-01220-f011:**
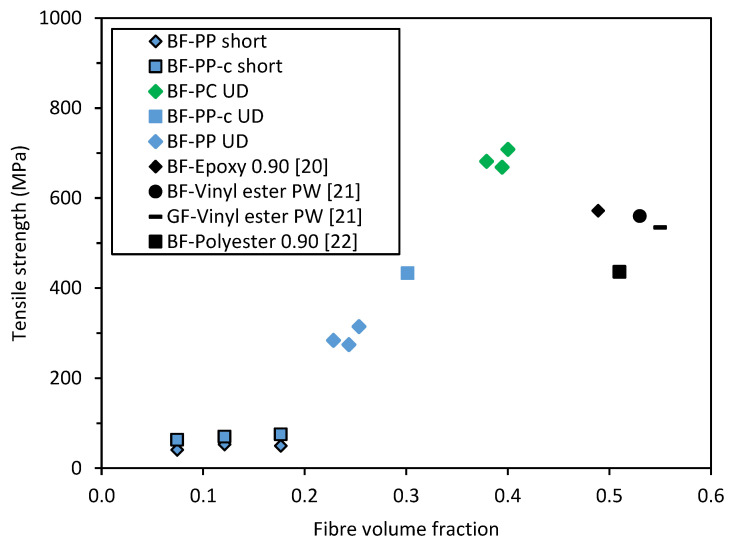
Benchmark of tensile strength values of BFRTP with other BFRP and GFRP. Note: Literature values are in black, and the experimental values are in green or blue. The sample nomenclature is by fibre type, matrix type and followed by the fibre architecture. For example, BF-PP short denotes the values for short BF-PP composites.

**Table 1 polymers-14-01220-t001:** Properties of basalt fibre with glass and carbon fibres [5,6].

	Basalt	E-Glass	S-Glass	Carbon
Density	2.63–3.05	2.54–2.62	2.46–2.54	1.78
Tensile strength	3000–4840	3100–3800	4020–4830	3500–6000
Elastic modulus (GPa)	79.3–110	72.5–78	83–91	230–600
Elongation at break	3.1–3.2	4.7	5.3–5.6	1.5–2.0
Maximum service temperature (°C)	650	380	300	500

**Table 2 polymers-14-01220-t002:** Processing parameters for BF-PC and BF-PP composites.

Matrix	Pre-Consolidation			Consolidation		
	Temperature (°C)	Time (min)	Pressure (bar)	Temperature (°C)	Time (min)	Pressure (bar)
PC	170 or 260 or 280	5	0.5	220	30	5
PP	170 or 210 or 250	2	0.5	160	30	2
PC, 10 bar	170	5	0.5	220	30	10
PC, 240 °C	170	5	0.5	240	30	5

**Table 3 polymers-14-01220-t003:** BF continuous fibre composites in this paper.

Matrix	Pre-Consolidation Temperature	Consolidation Parameters Variation	Notation Used
PC	170	-	PC-170
	260	-	PC-260
	280	-	PC-280
	170	240 °C	PC-170, 240
	170	10 bar	PC-170, 10 bar
PP	170	-	PP-170
	210	-	PP-210
	250	-	PP-250
PP with compatibiliser	170	-	PP-170-c

**Table 4 polymers-14-01220-t004:** Tensile and flexural properties of the UD basalt fibre PC composites.

Sample	v_F_	Tensile Strength (MPa)	Young’s Modulus (GPa)	Strain at Break (%)	Flexural Strength (MPa)	Flexural Modulus (GPa)
PC-170 (u)	0.39	669 ± 65	31.3 ± 0.90	2.21 ± 0.24	629 ± 108	19.1 ± 3.4
PC-170	0.38	n.d.	n.d.	n.d.	854 ± 45	27.6 ± 0.9
PC-260	0.40	708 ± 26	33.1 ± 1.8	2.30 ± 0.10	1011 ± 43	29.2 ± 1.8
PC-280	0.38	682 ± 20	32.1 ± 1.0	2.21 ± 0.05	923 ± 56	30.6 ± 1.6
PC-170, 10bar (u)	0.38	546 ± 19	30.0 ± 1.8	2.10 ± 0.13	624 ± 50	24.1 ± 1.1
PC-170, 240 °C (u)	0.37	523 ± 32	25.6 ± 1.0	2.13 ± 0.16	491 ± 62	21.7 ± 1.2

Note: v_F_ is the measured fibre volume fraction from TGA and n.d. indicated no data. “(u)” behind the sample label indicates the use of undried PC film in initial study.

**Table 5 polymers-14-01220-t005:** Tensile properties of the UD basalt fibre PP composites.

Sample	v_F_	Tensile Strength (MPa)	Young’s Modulus (GPa)	Strain at Break (%)
PP-170	0.23	284 ± 46	17.8 ± 1.1	1.98 ± 0.07
PP-210	0.24	275 ± 43	17.0 ± 1.5	2.46 ± 0.58
PP-250	0.25	315 ± 42	18.6 ± 2.0	1.87 ± 0.25
PP-170-c	0.30	433 ± 28	22.9 ± 2.9	2.09 ± 0.06

Note: v_F_ is the measured fibre volume fraction from TGA and values are within. Note that the annotation “-c” indicates that compatibiliser was used.

**Table 6 polymers-14-01220-t006:** Tensile and impact properties of the short BF-PP composites.

Fibre Loading (%) ^1^	Tensile Strength (MPa)	Young’s Modulus (GPa)	Strain at Break (%)	Impact Strength (kJ/m^2^)
0	35.8 ± 0.7	1.96 ± 0.49	292 ± 92	2.17 ± 0.21
20	40.8 ± 1.2	4.95 ± 0.43	4.82 ± 0.56	3.83 ± 0.35
30	52.8 ± 2.0	6.05 ± 0. 59	1.72 ± 0.25	4.98 ± 0.35
40	50.0 ± 2.9	9.29 ± 0.49	1.12 ± 0.14	4.81 ± 0.46
20-c	63.1 ± 2.1	5.80 ± 0.64	2.04 ± 0.16	5.38 ± 0.42
30-c	70.2 ± 3.4	8.11 ± 0.94	1.54 ± 0.13	7.39 ± 0.43
40-c	75.1 ± 3.5	7.73 ± 0.73	1.65 ± 0.23	8.49 ± 0.36

^1^ Intended fibre loading. The actual fibre loadings were measured by TGA and were within ±1% of the intended fibre loadings. Note that the annotation “-c” indicates that 2 wt% (w.r.t. fibre loading) compatibiliser were used.

**Table 7 polymers-14-01220-t007:** Parameters used for analytical modelling of Young’s modulus.

	Basalt Fibre ^1^	Polypropylene Polymer
Density (g/cm^3^)	2.8 [5,6]	0.90 [30]
Young’s modulus (GPa)	86.2 [5,6]	1.96
Length (µm)	249	-
Diameter (µm)	12.4	-

^1^ Average values from literature for the density and Young’s modulus were used, and the length and diameter of basalt fibre were measured using optical microscope and average values were used.

**Table 8 polymers-14-01220-t008:** Predicted and measured Young’s moduli of UD BF-PP and BF-PC composites.

Sample	v_F_	Experimental Young’s Modulus (GPa)	Theoretical Young’s Modulus (GPa)	Efficiency Factor ^1^ (%)
PP-170	0.23	17.8 ± 1.1	23.1	77
PP-210	0.24	17.0 ± 1.5	24.5	69
PP-250	0.25	18.6 ± 2.0	25.5	73
PP-170-c	0.30	22.9 ± 2.9	29.9	76
PC-170 (u)	0.39	31.3 ± 0.90	38.7	81
PC-260	0.40	33.0 ± 1.8	39.2	84
PC-280	0.38	32.1 ± 1.0	37.3	86

^1^ Efficiency factor is defined as the ratio of experimental Young’s modulus to theoretical Young’s modulus, expressed in percentage.

## Data Availability

The data presented in this study are available on request from the corresponding author.

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
