# Peer review of "Processing and Mechanical Properties of Basalt Fibre-Reinforced Thermoplastic Composites"

_polymers, 2022, doi:10.3390/polym14061220_

Round 1

Reviewer 1 Report

I think that the manuscript is interesting of a technical (engineering) nature, especially in the part concerning composites obtained by pressing.

However, I have a few criticisms:

In my opinion, optimization has not been performed in the work - as the title suggests. It is true that the influence of variable processing conditions on the properties of composites (processing temperature and additional pre-consolidation operation) was determined. However, no models describing the change of the properties of composites due to variable processing condition have been determined.

Please verify the use of the word optimization in the manuscript.

I advise against using abbreviations that were not explained earlier in the abstract - BF-PC, UD BF-PC.

The introduction should be greatly supplemented. The last paragraph is an abstract. There are no reports on the properties of composites with basalt, their mechanical and processing properties. The purpose of the research was not emphasized.

 V. 67. Maleic anhydrite polypropylene shall be Polypropylene-graft-maleic anhydride

V. 75. Please specify picture frame mold - shape, dimensions.

Vv. 126-129 Was the PP modulus determined at crosshed speed = 5 mm / min and for composites at 2 mm / min? In the case of testing polymer materials with viscoelastic properties, the determined modulus of elasticity depend on the rate of deformation.

Fig. 2 and Fig. 5 Why were the results approximated with a straight line? It seems that another relationship will be more appropriate (maybe a logarithm or a parabola?).

V. 231 Can the change in sample thickness be due to thermal shrinkage of the polymer? Higher polymer temperature = greater processing shrinkage.

For all results of mechanical properties, at least basic statistical analysis should be performed to determine the difference of means.

Table.7 Please explain why the difference in the value of the fiber length in the table with the given in material description (v.57 cut ... is aprx. 10 mm)

I recommend writing the variables in italics.

For UD basalt fibre, the simplest model described by equation 4 was used. It describes the maximum modulus under ideal assumptions. I recommend, although it is not necessary, to determine the simple efficiency factor for the these composites.

Author Response

Reviewer #1:

I think that the manuscript is interesting of a technical (engineering) nature, especially in the part concerning composites obtained by pressing.

However, I have a few criticisms:

In my opinion, optimization has not been performed in the work - as the title suggests. It is true that the influence of variable processing conditions on the properties of composites (processing temperature and additional pre-consolidation operation) was determined. However, no models describing the change of the properties of composites due to variable processing condition have been determined.

Please verify the use of the word optimization in the manuscript.

I advise against using abbreviations that were not explained earlier in the abstract - BF-PC, UD BF-PC.

Reply: We agree with the reviewer and have amended the title of the article from “Process Optimisation Study and Mechanical Properties of Basalt Fibre Reinforced Thermoplastic Composites” to “Processing and Mechanical Properties of Basalt Fibre Reinforced Thermoplastic Composites”. In terms of abbreviations used in the abstract, we have mentioned “BFRTP”, “PP” and “PC” earlier but we have not mentioned “UD” earlier in the article and have hence edited this in the revised manuscript. 

The introduction should be greatly supplemented. The last paragraph is an abstract. There are no reports on the properties of composites with basalt, their mechanical and processing properties. The purpose of the research was not emphasized.

Reply: As suggested, the introduction was substantiated.   

V 67. Maleic anhydrite polypropylene shall be Polypropylene-graft-maleic anhydride

V. 75. Please specify picture frame mold - shape, dimensions.

Reply: We have amended “Maleic anhydride polypropylene” to “Polypropylene-graft-maleic anhydride” and have specified the shape and dimension of the picture frame moulds used.

Vv. 126-129 Was the PP modulus determined at crosshead speed = 5 mm / min and for composites at 2 mm / min? In the case of testing polymer materials with viscoelastic properties, the determined modulus of elasticity depend on the rate of deformation.

Reply: Yes, the modulus of the PP and the continuous fibre composites were determined at cross head speed of 5 mm/min and 2 mm/min, respectively. ASTM 3039 states the cross head speed for composites (2mm/min) and for PP polymers, we have further taken reference with ASTM D638 which states in the fineprint that the cross head speed is to be selected for samples to break between 0.5 to 5 mins.

Fig. 2 and Fig. 5 Why were the results approximated with a straight line? It seems that another relationship will be more appropriate (maybe a logarithm or a parabola?).

Reply: We agree with the reviewer that the relationship need not be linear and have removed the linear trendline from Fig.2 and Fig. 5.

V. 231 Can the change in sample thickness be due to thermal shrinkage of the polymer? Higher polymer temperature = greater processing shrinkage.

For all results of mechanical properties, at least basic statistical analysis should be performed to determine the difference of means.

Reply: Slow and controlled cooling was employed during the compression moulding process and substantial thickness changes due to thermal shrinkage is not expected. Furthermore, minimal warpage was observed for the fabricated composites.

In accordance with the standards, at least 5 specimens were tested and the mechanical properties were within standard deviation of 20%, with most of the results within standard deviation of 10%. Unlike other analysis such as single fibre tests, which have substantially large standard deviation (sometimes even up to 50%), further statistical analysis is usually not expected to be performed.

Table.7 Please explain why the difference in the value of the fiber length in the table with the given in material description (v.57 cut ... is aprx. 10 mm)

I recommend writing the variables in italics.

Reply: The fibre length in Table 7 is much shorter than the cut length due to extensive shearing in the extrusion process that will reduce the fibre length. The cut length is the length of the fibre before the extrusion process and the fibre length in Table 7 is the measured length of the fibre after the extrusion process.

For UD basalt fibre, the simplest model described by equation 4 was used. It describes the maximum modulus under ideal assumptions. I recommend, although it is not necessary, to determine the simple efficiency factor for these composites.

Reply: The efficiency factor was added as a column in Table 8.

Reviewer 2 Report

Overall, the article is well designed and executed. But authors need to address following points before the acceptance of the article. 

  • Abstract need to be modified by adding (only) key results obtained from this work.
  • Introduction part is not sufficient. Need to add more information pertaining to materials and literature review.
  • Highlight contribution of the study to knowledge gap/specific problem.
  • Table 4 need to be explained in detail. Because it will help the reader in future.
  • “In addition, it was verified that the consolidation temperature of 220°C and pressure of 5 bar is the optimised consolidation parameters as varying the consolidation temperature and pressure did not yield better tensile properties”. Why? Please add proper reasoning.
  • Figure 4 needs more elaboration with proper reasoning.
  • Figure 7 needs some more SEM images to prove the claims.
  • Table 8 add one more column with respect difference between Experimental Young’s modulus and Theoretical Young’s modulus.
  • In the conclusion section only, the results are summarized. the author should be drawn together the most important results and their consequences.

Author Response

Reviewer #2:
Overall, the article is well designed and executed. But authors need to address following points before the acceptance of the article.

Abstract need to be modified by adding (only) key results obtained from this work.

Reply: We agree with the reviewer and as suggested, the abstract has been modified to be more concise.

Introduction part is not sufficient. Need to add more information pertaining to materials and literature review.

Highlight contribution of the study to knowledge gap/specific problem.

Reply: As suggested, the introduction was substantiated and the contribution of the study was also highlighted.   

Table 4 need to be explained in detail. Because it will help the reader in future.

“In addition, it was verified that the consolidation temperature of 220°C and pressure of 5 bar is the optimised consolidation parameters as varying the consolidation temperature and pressure did not yield better tensile properties”. Why? Please add proper reasoning.

Figure 4 needs more elaboration with proper reasoning.

Reply: As suggested, we attempted to add more reasoning and elaboration in the revised manuscript.

Figure 7 needs some more SEM images to prove the claims.

Reply: We agree with the reviewer and Figure 7 (a) and (b) were replaced with images containing more fibres to be more representative of the sample and they also substantiated the claims better.

Table 8 add one more column with respect difference between Experimental Young’s modulus and Theoretical Young’s modulus.

Reply: The difference between the experimental and theoretical values was added as the efficiency factor in Table 8.

Round 2

Reviewer 1 Report

After analyzing the presented manuscript, I conclude that the required corrections have been made. The manuscript may be published in presented form.

Author Response

Dear reviewer

Many thanks for your review.

Warmest Regards

Xinying